# Clustering of Small Territories Based on Axes of Inequality

**DOI:** 10.3390/ijerph19063359

**Published:** 2022-03-12

**Authors:** Xavier Perafita, Marc Saez

**Affiliations:** 1Observatori—Organisme Autònom de Salut Pública de la Diputació de Girona (Dipsalut), 17003 Girona, Spain; xperafita@dipsalut.cat; 2Research Group on Statistics, Econometrics and Health (GRECS), University of Girona, 17003 Girona, Spain; 3CIBER of Epidemiology and Public Health (CIBERESP), 28029 Madrid, Spain

**Keywords:** big data, clustering, hierarchical k-means, e-cohort, classifiers, machine learning, inequalities

## Abstract

Background: In the present paper, we conduct a study before creating an e-cohort for the design of the sample. This e-cohort had to enable the effective representation of the province of Girona to facilitate its study according to the axes of inequality. Methods: The territory under study is divided by municipalities, considering these different axes. The study consists of a comparison of 14 clustering algorithms, together with 3 data sets of municipal information to detect the grouping that was the most consistent. Prior to carrying out the clustering, a variable selection process was performed to discard those that were not useful. The comparison was carried out following two axes: results and graphical representation. Results: The intra-cluster results were also analyzed to observe the coherence of the grouping. Finally, we study the probability of belonging to a cluster, such as the one containing the county capital. Conclusions: This clustering can be the basis for working with a sample that is significant and representative of the territory.

## 1. Background

Currently, the concept of “health inequalities” refers to the impact that factors, such as wealth; education; employment; racial or ethnic group; exposure to environmental factors, including air pollution or weather variables; urban or rural residences; and/or the social conditions of an individual’s workplace or dwelling, have on the distribution of health and disease among the population. The study of the characteristics of the population and the geographical area of residence is the methodological support that allows for intervention points focused on the prevention and the disappearance of existing health inequalities to be identified.

Initially, socioeconomic inequalities were identified with health inequality [1]. Health inequality can be defined as an inequity in the spread of a disease. In other words, health inequality is the systematic and potentially avoidable differences in one or more health aspects across socially, economically, demographically, or geographically defined populations or population groups. Two conditions must be met for a difference in health to be considered as an inequality: (1) it must be considered socially unjust and (2) potentially avoidable (i.e., there are instruments available that could be used to avoid it) [1].

There is evidence that inequalities in health exist. While the Ladonde [2] and Black [3] Reports pointed this out, it was the Acheson Report [1] that firmly concluded that inequalities in health have a socioeconomic explanation. To date, twenty years later, most of these relationships have been demonstrated, and not an insignificant proportion is caused by environmental problems [1]. These factors are generally, but not exclusively, linked to gender, social and economic conditions [1,4,5].

In general, the living environment, and thus environmental conditions, can contribute to socioeconomic inequalities in health, either independently or, more likely, jointly [1,5]. The first is differential exposure: the most economically disadvantaged groups has a greater exposure to environmental problems, including, air pollution. The second is differential susceptibility to exposure (i.e., the main adverse health effects) resulting from environmental problems, which occur among the most economically disadvantaged people due to their greater vulnerability.

When we think about a longitudinal study to observe how health inequalities, individuals’ health, income, or another specific characteristic evolve over time, our thoughts very quickly turn to creating a cohort. This is immediately followed by considerations of the high cost and logistical difficulties of managing a cohort in terms of obtaining users, processing the sample, managing the information, and even handling and looking after the sample.

There are many cohorts in which the number of individuals easily surpasses 100,000 marks, including the Framingham Heart Study [6] the Current Management of Secondary Hyperparathyroidism: A Multicenter Observational Study (COSMOS) [7], and the NutriNet-Santé Study [8]. When the sample is large, the governance of the user and their data become extremely costly. The sample is acquired in the traditional way, via a letter explaining to the individual concerned that they have been selected to take part in a project and what it consists of involves some costs that are sufficiently high as to consider alternatives to the cohort [9,10,11,12]. Another point of consideration is that the cost of increasing, improving, or simply demonstrating the significance for a group or subgroup that was not initially contemplated can be so high that many researchers decide not to incorporate any more individuals into the cohort beyond a theoretical framework. Financial constraints and a lack of logistical resources are factors that generally mean that traditional cohorts have limits.

This is where digital considerations come into play. An *electronic-cohort* or an *e-cohort* is a traditional but digitally managed cohort [13]. This management can be entirely digital via user interactions with websites, platforms, apps, or by post [9]. It can also be of a hybrid nature, depending on the type of information needed to be previously collected and the level of difficulty of obtaining the information automatically. Some traditional cohorts, some of them *novel cohorts* with a high number of individuals, are starting to test the transformation of traditional cohorts into electronic cohorts, seeking their improvement. These improvements basically focus on optimizing the cost/efficiency of the project and obtaining and managing data.

The marginal cost of the sample in an e-cohort is practically zero [11], although some costs inherent to longitudinal studies and linked to maintaining and managing the sample remain. They are, nonetheless, significantly lower than the cost of traditional acquisition. This cost reduction not only signifies monetary savings, but also logistical ease in terms of the human factor. Currently, the e-cohorts that have published results focus on using a webapp as the working platform, sometimes including external elements, such as smartwatches [14] or diaries that must be kept up [9], with the user being able to choose different format. These external elements end up not being used by the individuals, causing sample mortality and making this a weakness of e-cohorts that needs to be addressed [10,11,14] to be able to obtain data without the user having to directly intervene with the app or the mobile phone.

The e-cohort also reduces the costs linked to data collection, minimizing the logistical costs of obtaining, cleaning, homogenizing, processing, and automating all the information concerning the sample. In a cohort, the time spent purging everyone’s information quickly adds up to many hours, while digitally doing so allows for “interviewing” the sample, thus eliminating the time spent on this task. We must also consider that the information is obtained in this way just once or twice a year, especially if the sample is large. This lack of information about the user during certain periods causes a data lag, generating an information gap that the traditional cohort cannot resolve. The e-cohort enables different and several surveys to be carried out at no extra economic cost, although consideration must be given to ensure that the sample is not saturated with activity.

In e-cohorts, the data can be obtained in different ways, which, for the sake of simplification, can be separated into two groups: the first where the user interacts, and the second where the user is “passive”. In the first, the user interacts directly with the website, app, or mobile device, and consciously responds to the information requested, such as answering a survey or a question about their perceived state of health. Although users’ fatigue thresholds have not yet been established, the e-cohort is an attractive option, thanks to the possibility of asking more users more questions at a lower cost. In addition, all the answers enter a digital process where they are easily automated, further reducing the cost and increasing the efficiency of the process. The same logic can be applied to the use of external elements, for example, a smartwatch that can supply minute-by-minute information about the evolution of an individual’s heart rate. The results obtained using these tools are unbiased compared to the data obtained using traditional tools, and they also provide information that is consistent over time.

It has been demonstrated that the most effective way to gather users for a sample is by offering a monetary incentive [9,12,13], which the user receives once they have responded to the questions.

There has been a case in which the sample was opened up by applying citizen science. In these cases, the e-cohorts have to buy their sample with a census, or via a similar means, to validate whether the sample obtained is representative of the study population [11,13]. The sample must be validated by separating the different demographic characteristics. In various cases, it has been observed that there are groups that do not tend to take part in these experiences, so additional efforts are required to sample these groups correctly. Conversely, young women with a higher educational level tend to participate most in this type of initiative, leading to their oversampling [14]. This can cause biases, which must be controlled when performing the inferences. It has also been shown that a population with little or no digital skills find responding to the questions problematic. Despite this limitation, very few individuals emerge to complicate the sampling of specific groups [11].

One common limitation of the cohorts that is not resolved by the e-cohort emerges when seeking a way to use a sample to represent a set of territories. If we want to significantly represent the population of Catalonia, it is sufficient that it is random throughout the territory. Meanwhile, if we want to work with a specific axis, such as age, it is sufficient to make a small adjustment and increase the size of the sample.

The Public Health Observatory of Girona Province (Dipsalut) is designing an e-cohort to carry out a longitudinal study to simultaneously examine the health of the population and its socioeconomic situation. The province of Girona is defined as a semi-rural territory [15], with 221 municipalities and a population of approximately 770,000 people. Less than 10% of the municipalities have more than 10,000 inhabitants, substantially limiting statistical significance and causing us to encounter the limitations of the statistical secret.

This e-cohort must not only allow us to obtain a significant representation of all the municipalities in the territory, but it must also optimize the resources and the sample. A municipality codified as LAU level 2 by Eurostat is the smallest existing territorial division at the national level in Spain, where there is a decision-making power over local policies. The present paper explains the process of carrying out clustering in the province of Girona. The clustering must allow similar municipalities to be clustered for the purpose of constructing a representative sample of the different territories. This sample must enable the generation of a set of indicators that present the inequalities that exist in the territories [16]. Furthermore, its design must revolve around the five major axes of inequality: sex, age, social class, migratory process, and territory. This sample was controlled and had to be regulated, so working with an open sample was not a consideration.

This paper explains the process used to cluster the municipalities into 6 groups according to their similarities, and how 14 clustering algorithms were tested to find the ones were the most effective and representative of the province. Finally, statistical modeling was used to observe if there were significant differences between the clusters to draw the final conclusions.

## 2. Methods

### 2.1. Methods Prior to Carrying out the Study, the Data Set, and the Data Sources

As explained earlier, the diversity of the territory of Girona requires a large number of variables to determine the differences and similarities between its municipalities. These differences can range from an economic point of view, where the main cities in the province have a larger number of specific companies and sectors, to the migratory processes that the areas experience or the number of elderly people who live there. As can be seen in Figure 1, we carried out a review of all the indicators that exist in the main databases that provide information on the municipalities in the province of Girona. From this, we obtained 541 variables. These were then processed based on the availability of data for the study period, data availability for most municipalities throughout the study period, as well as the elimination of variables that we considered to be duplicates or redundant, and those that did not contribute any relevant information to the study.

Prior to the clustering, a final set of 54 potential variables encompassing the areas of demography, economy, job market, public spending, health, and populational and geographical incidences and emergencies were identified.

#### 2.1.1. Data Sources

The data used were supplied by different official sources. They were the Statistical Institute of Catalonia (IDESCAT) [17,18,19,20,21,22,23], Xifra [24,25,26,27,28,29,30], Open Data Generalitat [31,32,33,34], the Department of Territory and Sustainability of the Government of Catalonia [35,36], and the National Statistics Institute (INE) [37]. Obtaining information for 199 of the 221 municipalities was difficult because many of them are bound by the obligation of the statistical secret due to the small population figures of below 10,000. 

#### 2.1.2. Demographic Area

Ethnic and cultural diversity and populational polarization have positive repercussions on the economy and generate cultural and social combinations [38]. Migratory movements also have an effect on the socioeconomic levels of the population [39], causing modifications to the diseases and states of health linked to the populational pyramid that can lead to changes in health policies.

The following indicators were used to evaluate the demographic situation of each municipality: the average age of the population, total population, population resident abroad [23], net migration and population [22], immigration rate and the native population index [27], and population density [30].

#### 2.1.3. Economic Area

Economic capacities can determine the significant differences between the inhabitants of a municipality. As described in the literature [40], poverty does not solely consist of the economic capacity of a person to meet minimum expenses, but it also has implications in terms of health, education, and the chance to save money to have a better quality of life. The standard of living can also be determined by access to basic goods, such as housing.

The following indicators were used to evaluate the economic status of each municipality: personal income tax [20], the result of the tax return per declarant [28], and gross income per person [37]. The following types of indicators were collected to evaluate the degree of poverty in each municipality: the distribution of the sources of income and the Gini index [24].

The state of housing was also included as an economic indicator, because a direct relation between the state of housing and the economy of a municipality is considered to exist, including the number of residences, average rental price [40], cadastral value and number of urban plots, and number of immovable properties and their cadastral value [24].

#### 2.1.4. The Job Market Area

A municipality’s job market shows the type of employment that exists in that area and the predominant sector. Depending on the sector, the industry and the working conditions linked to the different sectors of a municipality, the lifestyle of the people that live there, are positively affected to varying degrees [41].

The following indicators were used to evaluate the job market of each municipality: social security affiliations, according to the registered home address of the affiliated person and the activity sector; social security affiliations according to the percentage of the active foreign-born population [17]; unemployment [26]; unemployment among foreign-born persons [26]; and the temporary employment rate [25].

#### 2.1.5. Area of Public Spending

Public spending shows the amount of money spent by the local government of a municipality to cover the needs of its inhabitants. There is discussion in the literature as to whether an increase in public spending has a direct impact on citizens and their levels of poverty [42,43,44], health [45], and education [46]. 

The following indicators were used to evaluate the public spending of each municipality: the number of libraries [18] and sports facilities [32].

#### 2.1.6. Area of Health

This area considers the state of health of the inhabitants of a municipality. Given that the territories were generally very small, we had access to data that were more purely biological. Traffic accidents were also observed as they impact the health of a territory and its preventive strategies, focusing on pedestrians, cyclists, cars, and motorcycles [47]. Aging must also be considered in this area, since it is one of the most predominant demographic phenomena in Europe in the twenty-first century. There are indexes that show how aging has different effects on the population in terms of fertility, age, and birth rate [48]. This phenomenon involves some specific public policies that have a direct impact on the population and their state of health.

The following indicators were used to evaluate the state of health of each municipality: the number of births and deaths and the gross mortality rate [19] and birth rate [29]; the number of traffic victims to evaluate the possible impacts on the inhabitants of a municipality [49]; the variables of the aging and global dependency indexes [30]; and the Synthetic Fertility Index and the natural population growth [29].

#### 2.1.7. Area of Population Incidences and Emergences

The incidences and emergencies of the inhabitants of each municipality show the population’s one-off and recurrent needs in terms of the emergency services. There are social factors that generally contribute to the use of these services [50]. The following indicator was used to evaluate the incidences in each municipality: the number of emergency phone calls [31].

#### 2.1.8. Geographic Area

The geography of the province of Girona is diverse and varied. There are coastal, mountainous, and flat areas, and the geographical characteristics of each area is instrumental in the development of a type of commerce and populational structure. The following indicators were used to evaluate the geography of each municipality: the extension of herbaceous crops [34] and woody cultivation [34], land extension in km^2^, and the singular entities in each municipality [21]. The altitude, latitude, and longitude of each municipality were added later [21], in addition to whether it was a county capital. Additionally, included was whether these municipalities were in a mountainous area [36] or coastal [35]. These variables show the different types of environments and their geographical positions.

#### 2.1.9. Alternative Data Sets 

Two databases parallel to the working one were developed: a nominal data set and a smoothed data set. These had to enable the observation of whether the smoothing of data or the transformation of the indicators from a percentual to a nominal value improved the cluster forming. In the nominal data set, the data was obtained from the sources mentioned above. A z-score transformation was performed for the smoothed data set [51]. The same number of variables was maintained in both datasets.

### 2.2. Control of Missing Value or Statistical Confidentiality

There was a set of data that was lost because they are bound by the obligation of the statistical secret, so they could not be collected. In these cases, an estimated value was assigned to each of those lost sets.

### 2.3. Variable Selection

We carried out a variable selection process, spike and slab, according to the population [52]. The aim was to eliminate the redundant variables and excessive noise. Other methods for selecting variables were also employed: Ridge Regression [53,54], LASSO [55], Elastic Net [56], SCAD [57], MCP [58] and LARS [59]. 

### 2.4. Cluster Analysis

Once the variables were selected, a clustering process was carried out to detect the municipalities that were similar among them. Given that the data set represented such different types of municipalities, it was decided to carry out a preliminary task with 14 different algorithms. This process was required to observe the algorithms that adapted best to the type of data, which is why they were of different types: partitional, hierarchical, one-pass, density-based, and big data clustering.

Among the partitional clustering methods, the following were used: k-means [60], Partitioning Around Method (PAM) [61], Clustering Large Applications (CLARA) [61], fuzzy clustering [61], CLARANS [62], and EA [63]. The hierarchical methods employed were Divisive Analysis (DIANA) [61], Agglomerative Nesting (AGNES) [61], and hierarchical k-means [64]. Additionally, the density-based methods used were SUBCLU [65], DBSCAN [66], and OPTICS [66]. The big data and One Pass methods were BICO [67] and BIRCH [68].

The cluster analyzed responds to a grouping based on a measure of distance where each observation initially acts as a cluster.
$$X=\left\{x_{i}|i=1,\ldots ,\;r\right\}\;\mathrm{com}\;a\;\mathrm{dades}\;\mathrm{base}\;A$$
$$A=\left\{a_{i}|i=1,\ldots ,n\right\}\;\mathrm{for}\;n=53$$

These clusters fuse iteratively together, depending on their proximity until no more of them can be fused.
$$Ci=\left\{c_{ij}|j=1,\ldots ,K\right\}\;\mathrm{for}\;K=6$$

Each new fusion can generate a new centroid in each cluster.
$$D=\left\{d_{i}|i=1,\ldots ,K\right\}\;\mathrm{for}\;K=6$$

#### Mapping of the Clustering

The clusterings created using the hierarchical k-means algorithm were represented to evaluate whether they followed a geographical pattern on the map of the region under study (i.e., Girona). The map was created for three points in time, 2015, 2016, and 2017. The maps of the municipalities were obtained from the Cartographic and Geographic Institute of Catalonia [69]. The mapping was also used to observe whether there was a variation in the municipalities over the years.

### 2.5. Data Analysis

A multinominal logistic regression was carried out, for which the dependent variable ($\pi_{j}$) is the cluster generated, where *j* = 1, 2, 3, 4, 5. The variable of the reference group was 6, modeled in the following way:$$logit\left(\pi_{j}\right)=\log\left(\frac{\pi_{j}}{\pi_{6}}\right)=x^{T}\beta_{j},\;\mathrm{for}\;j=1,\ldots ,5$$

It was adjusted as follows to find the estimated probability (${\hat{\pi }}_{j}$) of the events:$${\hat{\pi }}_{6}=\frac{1}{1+\sum_{j=2}^{j=6}exp\left(x^{T}{\hat{\beta }}_{j}\right)},\;\mathrm{for}\;j=6$$
$${\hat{\pi }}_{j}=\frac{exp\left(x^{T}{\hat{\beta }}_{j}\right)}{1+\sum_{j=2}^{j}exp\left(x^{T}{\hat{\beta }}_{j}\right)},\;\mathrm{for}\;j=1,\ldots ,5$$

The final result enables the clusters to be compared with the municipality of Girona.

### 2.6. Software

All the analyses were carried out using the free R software. The packages used were *glmnet*, *ncvreg*, *lars*, *spikeslab*, and *data sets* for the variable selection method; *data sets*, *stats*, *factoextra*, *cluster*, *dbscan*, *subspace*, *stream*, *clv*, *stream*, and *fpc* for the clustering and validation of the clusters; *nlme*, *tidyverse*, *moments*, and *nnet* for mining the data; and *factoextra*, *ggplot2*, *gridExtra*, *cowplot*, *rgdal*, and *tmap* for the graphic representation.

## 3. Results

### 3.1. Area and Period of Study

A process of clustering small areas of Catalonia using a set of 54 variables was carried out. A prior task was performed to select the variables that were most relevant to the different areas, as explained in the following sections.

The study period was initially 2010 to 2018. However, given the small dimensions of both the territory and population, the data are bound by the obligations of the statistical secret, presenting limitations regarding accessing the available information. Consequently, the study period was changed to 2015–2017, when the data are more consistent and relatively unproblematic regarding lost values. All the municipalities were therefore represented by a high level of consistency.

In this study, we considered 221 of the 948 municipalities belonging to the region of Catalonia. The number of inhabitants varied between 83 and 99,013 (average inhabitants: 3412, standard deviation: 9081.349 inhabitants, median inhabitants: 746, Q1 298 inhabitants, and Q3 2290 inhabitants). The population density varied between 1 and 4493 inhabitants per km^2^ (average: 45 inhabitants/km^2^, standard deviation: 464.216 inhabitants/km^2^, median inhabitants: 45/km^2^, Q1 20 inhabitants/km^2^, and Q3 130 population/km^2^).

### 3.2. Variable Selection

To eliminate the redundant variables and excessive noise, we carried out a variable selection process, spike and slab, according to the population [52]. The models were based on the relationship with respect to the number of inhabitants in a municipality. The mean squared error of the predictions was used as a method comparison criterion [70]. The spike and slab method presents the smallest mean squared error (MSE) (see Table 1).

The dimensions of the final dataset are defined in 54 variables for 221 municipalities over 3 years, thus obtaining a final sample of 35,802 cases.

### 3.3. Clustering

The number of clusters obtained from the supervised methods was six (Figure 2). This number was validated based on the application of the Elbow method in a task carried out prior to the process of clustering. The number of optimized clusters does not change in any of the three data sets.

The results of the clustering process are presented in Table 2 (external and internal validation of clustering), Table 3 (number of observations for each cluster and data set), and Figure 3 and Figure 4 (results of clustering).

The diversity of the municipalities in Girona presents a well-recognized heterogeneity. The capital has a little over 100,000 inhabitants (103,369 inhabitants), while there are less than 50,000 (47,235 inhabitants) in the next largest municipality. There is also important diversity in a geographical sense, with a set of municipalities located in mountainous areas and others located on the Mediterranean coast. This heterogeneity across the entire area generates some obvious socioeconomic and health differences. The density-based clustering algorithms do not work this heterogeneity optimally. Many municipalities, including the capital of the province, are detected as outliers. This type of algorithm does not allow all the municipalities to be classified, and so they were ruled out. However, the rest of the models classified all the municipalities (see Figure 3).

An external and internal validation study was carried out to choose between the rest of the algorithms. A graphic validation was later designed using a cloud of points and the mapping of the clusters. The clustering produced by the hierarchical k-means method was consequently chosen.

As shown in Table 2, the internal validation values [71] of the algorithms, k-means, hierarchical k-means, PAM, and CLARANS, present the optimum values in the original database. In the nominal and smoothed data set, we observe how the PAM algorithm obtains some internal validation results that are inferior to the rest of the previously mentioned algorithms.

The external validation shows how PAM is the algorithm presenting a difference between inferior clusters in all the data sets. However, the intra-cluster difference varies depending on the data set. The three algorithms that present the relation of the most optimum intra-between cluster differences can be highlighted: k-means, PAM, and hierarchical k-means. The entropy value [71] that shows the best clustering is presented in the fuzzy, DIANA, and AGNES algorithms for the different data sets. The CH index [72] shows how the k-means, PAM, CLARANS, and hierarchical k-means algorithms are the ones that present the best construction of the clusters.

Table 3, which shows the distribution of the clusters, helps with the conceptualization of the dimensions of the clusters. It can be observed how the different clustering has a main cluster in the original data set, which has a greater number of cases than the rest. This main cluster varies from 186 to 627 in the different algorithms. There are two types of clustering: those in which the main cluster captures most cases, and those in which the cases are distributed more homogeneously between the clusters. In most of the groupings, there is a second cluster with a weight greater than 20% for all the observations. The groupings in which the main cluster retains at least 50% of the sample are CLARA, CLARANS, hierarchical k-means, fuzzy, BICO, EA, DIANA, and AGNES. Meanwhile, k-means, PAM, and BIRCH are the algorithms that distribute the individuals in the most balanced way. The nominal and smoother data sets present a more uniform distribution of the clusters in the municipalities.

Once the validations of the clusters and their dimensions have been analyzed, a graphic representation of them must be produced. This representation must allow the algorithms that generate a visually intuitive clustering to be detected to facilitate choosing the final clustering (Figure 3).

Figure 4 shows how the k-means, PAM, and hierarchical k-means algorithms are the dimensions that generate a more visually intuitive clustering for the different data sets. The representations based on the nominal data set show how the distribution is reduced. In the smoothed data set, the cases are smoothed in a more obvious manner.

The graphic representation using the clouds of points does not allow a pattern that is significantly better than the rest to be detected. Therefore, Figure 5 shows the groupings of the k-means, PAM, and hierarchical k-means algorithms on the study map (province of Girona).

### 3.4. Mapping of the Clustering

The maps illustrate how the clustering carried out using the original data set enables us to detect that the k-means and hierarchical k-means algorithms differentiate between the set of coastal municipalities and some county capitals together. They also cluster the set of inland municipalities that link Barcelona and France. They do not detect a differentiation between the mountain municipalities, although they do differentiate between a subregion of them. A small cluster for some of the municipalities with a high population is generated. Regarding PAM, the mountain and coastal municipalities are clearly differentiated. Some county capitals are also added to these last clusterings. A set of municipalities very close to Barcelona and the municipalities nearest the French border can be identified, as can the inland municipalities dispersed in a first and second ring around the county capitals. In all three clusterings, Girona is grouped independently.

The clusters generated by the k-means, PAM, and hierarchical k-means algorithms, based on the nominal and smoothed data sets, are very similar. The k-means and hierarchical k-means algorithms detect the first grouping of the municipalities located in the mountainous areas. K-means detects a subset of these municipalities since they belong to the inland municipalities. Both algorithms also detect a set of municipalities that belong to the coast, together with some county capitals. The municipalities nearest the French border and those closest to Barcelona are detected. Meanwhile, PAM detects a pattern among the municipalities next to France (Figure 6 and Figure 7).

### 3.5. Descriptive Study of the Clustering

Table 4 shows the variability of the clusterings. Notably, the k-means and hierarchical k-means algorithms are the data sets with the least variability in all three data sets, indicating that these clusterings do not undergo changes and are stable over time.

The algorithm chosen is hierarchical k-means, because it presents the optimum and secure properties to generate a sample that endures over the years. Six clusters can be detected in this algorithm. The first cluster contains the municipalities near the French border (Empordà), and the second contains the municipalities located in mountainous areas. The third group focuses on the inland municipalities of the territory. The fourth group is made up of the coastal municipalities and some provinces in the county. The fifth group detects the territory’s important municipalities, be it economically or in terms of population. The sixth and last group separates the capital from the rest of the municipalities.

The results of the descriptive study of the clustering are shown in Table 5 (descriptive analysis by conglomerates, robust values). As can be observed, the size of the population is very different among the six groups. There is an obvious contrast between the high number of people that live in the capital (98,255) and the median population of the municipalities located in the other county capitals (37,042) and close to the coast (10,709), with lower population numbers than the rest of the cluster. The population density is also higher in these groups, and especially in the capital (2512). It can be observed how the native population figures are quite similar for all the clusters, except the capital, where this figure is higher (40.22). Meanwhile, the ratios of immigrants in the inland municipalities (0.082) and the mountainous areas (0.061) are lower than in the rest of the clusters, with the highest ratios in the coastal municipalities (0.217) and the other county capitals (0.225).

The internal and external flow of movements is greatest in the capitals of the county (28) and in the capital of the province (555). The migratory balance is also higher in the capital than in the rest of the clusters. The different weights in the distribution of jobs in the sectors in each cluster can also be observed. The mountain and border clusters (7.23 and 6.61, respectively) have the highest percentage of the population employed in agriculture. Meanwhile, the inland municipalities (20.98) have a higher percentage of the population employed in the industrial sector. The weight of the construction sector is similar in all the clusters, except for the capital, which has a lower percentage (4.67). The services sector predominates in all the clusters, with the greatest weight (81.94) in the capital. The unemployment rate increases in line with the weight of the population of each cluster. Likewise, the clusters with the highest population densities are where the Gini index is highest. 

Inequality is greatest in the capital (36), followed by the coastal municipalities (34.60) and the main county capitals (34.10).

Income from salaries is highest in the capital (10,277) and lower in the coastal municipalities (7393) and the county capitals (7218). Income derived from unemployment benefits is lower in the coastal municipalities (2488) and the county capitals (2221).

The cost of renting housing is similar among the clusters. However, the cadastral value is not, with the highest values in the capital (4,005,166) and the lowest around the French border (22,797.5).

On observing the breakdown of the population balance, it can be observed how this balance is lower in the mountainous areas (1) and the border areas (2), than in the capital (704) and some other municipalities (193). A similar dynamic appears in relation to the natural growth of the population and the dependency index. The border and mountain municipalities have the same negative natural growth rate (−1) and the highest dependency indexes (60.60 and 56.19). Conversely, there is a higher natural growth rate in the capital and county capitals and a lower dependency index (50 and 48.04). The number of traffic accident victims is similar in all the clusters, except in the capital (76). However, more phone calls are registered in the coastal municipalities (5) and the other county capitals (5) than in the rest of the clusters. 

Geographically, it can be observed how the highest municipalities are found in the mountain municipalities (953.5).

### 3.6. Inference

The clusters represent the variability of the territory, which, as we have shown, is very varied, and therefore these different realities are so different that they do not follow a normal distribution. The Kruskal–Wallis [73] and Mann–Whitney tests [74] show that there are significant differences among the clusters. To observe these differences from the clusters, we assume that we do not have the presence of multiculturalism or outliers. A multinominal logistic regression was performed to observe these differences [75]. The odds ratios of the regression are presented in Table 6.

## 4. Discussion

The execution of the algorithms and data sets show how the validation improves when working with more stable data, such as the nominal values smoothed by the z score. This stability is translated into less variability in the construction of the clusters in the three periods. The variability improves when working with the smoothed data set. This is a relevant point when considering the design of a longitudinal study to find the individuals that are representative of the same type of municipality.

Of the clustering presented, three of the maps can be identified as the most representative of the territory. The first was the map created with the original data set using the PAM algorithm, which managed to determine six clusters: the French border, the mountainous area, the outskirts of Barcelona, the coast, the area inland, and the capital of the province. The other two were the maps generated with the nominal and smoothed data sets, using the hierarchical k-means algorithm, which showed five clusters: the French border, the mountainous area, the coast, the area inland, and the capital of the province, in addition to a sub-cluster of the main county capitals. The more solid algorithm was chosen at the expense of the loss of the cluster adjoining Barcelona.

The multinominal logistic regression shows that there are differences among the clusters and the capital. There are no significant differences demographically between the municipalities grouped as county capitals and the capital of the province. The clusters of the mountainous areas, the French border, and the coast have the probability of having lower population balances and lower population densities than the capital. Consequently, the probability of having a higher global dependency index than the capital is higher. 

Economically, there are less differences between the clusters. Any differences stem from salaries with respect to the capital, giving the coastal areas a lower probability. However, they have a higher probability of obtaining a gross average income and a pension than the capital. On the coast, both the gross average income and income from salaries have a higher probability of being the same as those of the capital. However, pensions have a lower probability.

The probability of having a rental housing offer equal to that of the capital is less in the mountain, border, and coastal clusters. However, the probability of owning property is higher with respect to the capital. Nonetheless, there is less probability that they are valued the same as the capital.

The job market presents significant differences, except for the municipalities in the county capitals. In the rest of the clusters, there is a greater probability of being unemployed than in the capital. Notably, the probability of having an immigrant unemployment rate equal to that of the capital is lower on the coast and in the mountains. The probability of having workers who are employed in the agricultural sector with respect to the probability of the same in the capital is greater in the coastal municipalities, and lower in the other municipalities. The probability of having workers employed in the services sector works inversely.

The probability of having sports facilities and libraries with respect to the capital is higher in the coastal municipalities. We find the inverse in the border municipalities, which have a negative probability. There are no significant differences in the municipalities of the county capitals.

In terms of interpreting the health variables, there are no significant differences with respect to the municipalities of the country capitals. For the rest of the clusters, the probability of having an aging index, similar to that of the capital, is negative. The probability of having the same death rate as the capital is higher in the border municipalities and on the coast, and negative in the others. The recovery index also has a negative probability. In the inland municipalities, the probability of having a mean age equal to that of the capital is lower than in the rest of the clusters. Traffic incidences and victims are more probable in the mountain and coastal municipalities than in those of the capital.

Clear differences were observed between the clusters and the capital. However, few significant differences were observed in the subgroup of the municipalities in the county capitals.

In conclusion, working with microdata is complicated, in terms of both making comparisons and modeling and clustering, especially if they are socioeconomic data. The difficulties of working with indicators, indexes, and rates complicate the data mining process and, later, the reading of the results. A smoothing or standardization process is necessary to work effectively. It must be considered that using percentages with such small data sets mean that these can drastically change from year to year. These possible irregularities accentuate the variations and generate an elevated volatility. This volatility affects the clustering and models, making their classification difficult. These factors end up translating into a high variability of the observations in the groups. However, this way of working can end up impeding the detection of new emerging clusters.

The functions based on density do not work optimally with variables that have such different realties as these. Figure 3 shows how they do not manage to classify all the municipalities. It should be tested whether re-clustering the outliers results in being able to classify all the municipalities, even though this means generating a final clustering superior to the k-number of the chosen clusters. The hierarchical k-means and k-means algorithms generate a cluster that does not present large significant differences with respect to the capital, so we can therefore work with five clusters rather than six. This helps us to design the simplest sample with the possibility of generating the most segregations. Another point for further study is whether the subgroup detected by PAM presents significant differences to the other groups to maintain the six clusters. A priority when designing a clustering to be able to extract a set of individuals to carry out a longitudinal study using digital tools is that these groupings endure for as long as possible. 

Another point to bear in mind is that the number of years studied should always be higher than the number of clusters we want to create. This way, we can know in which cluster the municipalities are classified, most times, to be able to find a cluster–territory relationship and a trend. This was not possible in this study due to the lack of data.

## 5. Conclusions

This article aims to help researchers and other decision-making institutions facilitate a comparison of the structuring and grouping of small areas, especially in those cases where the differences between them are so large. It also endeavors to show an optimal way of transforming and working on datasets to facilitate the resulting groupings. Two of the main limitations in grouping such diverse and small populations is, on the one hand, the lack of data and, on the other, the lack of experiences that endured over time, where we can observe their evolution.

If we want to analyze the impacts of spatial variables such, as NDVI or the pollutants PM_2.5_, PM_10_, NO_2_ or CO_2_, it is advisable to generate data at a lower level than the municipality, as municipalities, while not the smallest administrative division, are the smallest division that has political decision-making power. This would allow us to segment a cohort from census tracts or districts in the future and reduce the potential ecological fallacies that cohort data may generate. In addition, it would capture the inequality that can be observed between the rich and poor areas in cities better. The lack of experience working in small areas, along with the nature of most indicators, makes these processes difficult.

Currently, it is essential to start generating data at the scale of small areas, even smaller than those of a municipality, because otherwise we will not always be masking the inequalities through averages and aggregate values of population subsets, in which wealth has blurred the levels of poverty. On the other hand, the microdata permits the creation and adaptation of new indicators that allow the inequalities and the phenomena that occur in the territorial field to be captured more efficiently.

Data protection policies, although necessary, often prevent the study of the reality of territories. They also make it difficult to study individuals in a particular way. These mechanisms end up making it difficult to observe inequalities as well as study the sensitivity that each individual has, with respect to their social conditions and how these affect them.

To facilitate the best clustering process, it would be useful to carry out trend studies and predictive modeling to observe the subsequent years and to be able to forecast where each municipality will be classified, to help create a clustering that endures over time. 

## Figures and Tables

**Figure 1 ijerph-19-03359-f001:**
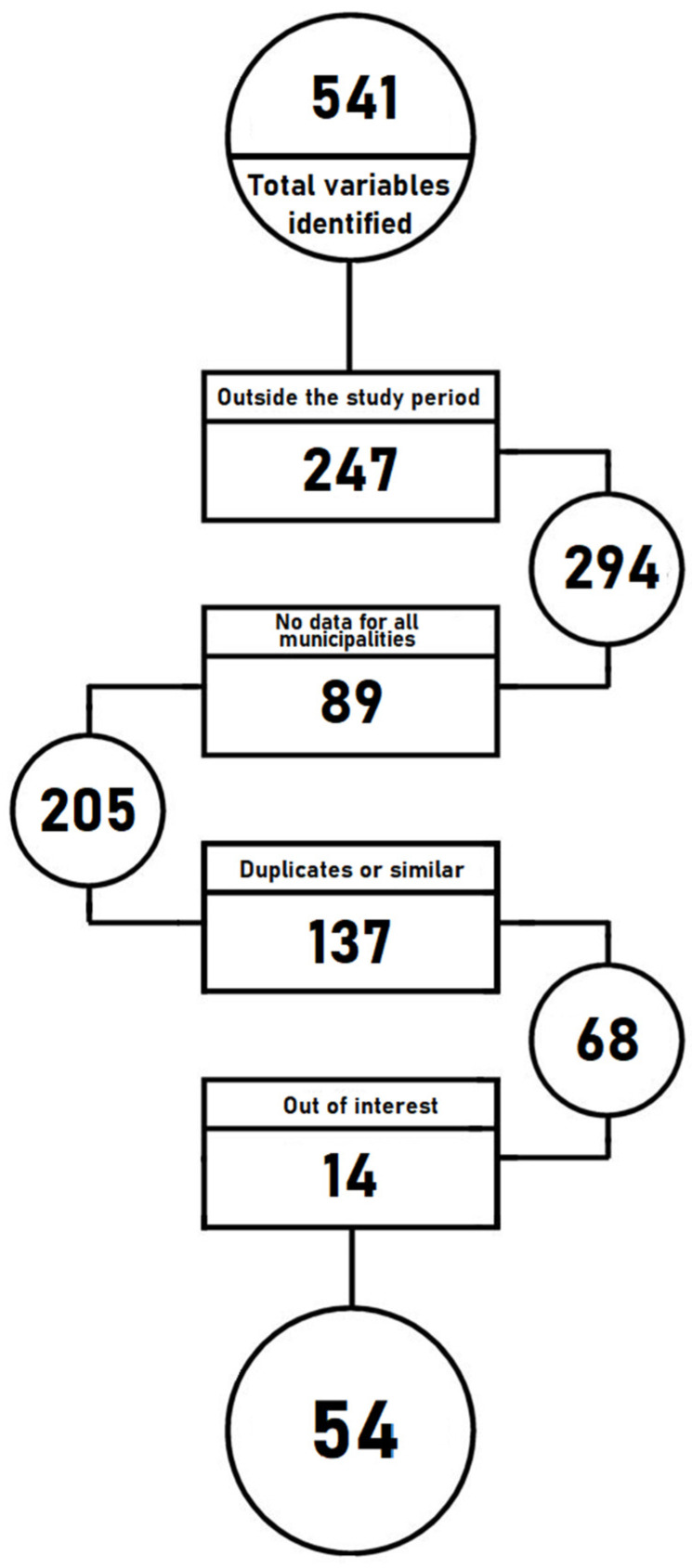
Debugging the process of all the detected variables up to the final data model. Source: authors’ own elaboration.

**Figure 2 ijerph-19-03359-f002:**
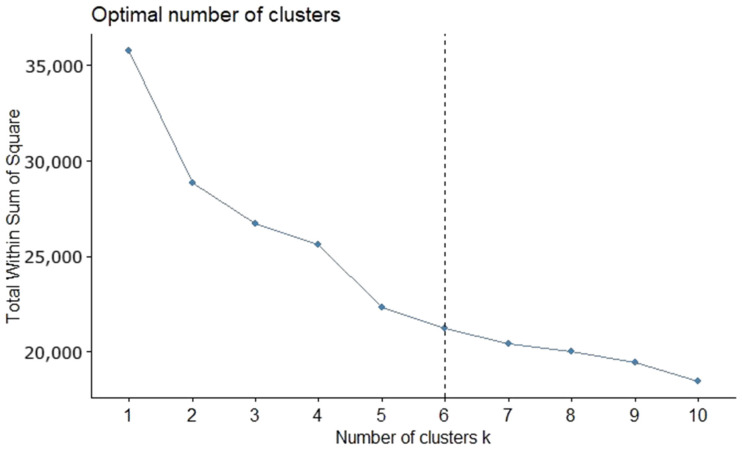
Process of obtaining the optimal number of clusters from the Elbow method. Source: authors’ own elaboration.

**Figure 3 ijerph-19-03359-f003:**
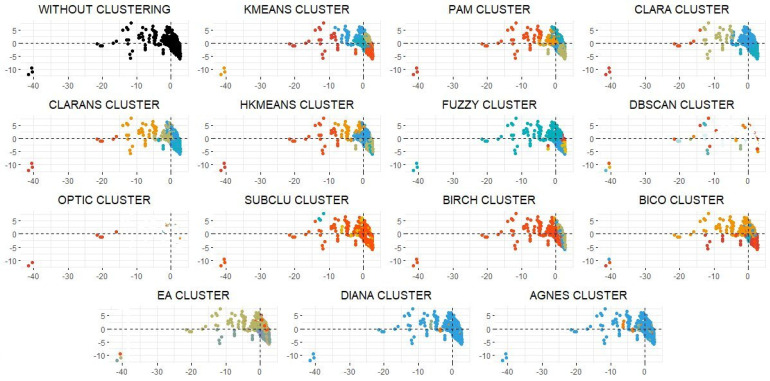
Representation of the different algorithms performed to study the clustering of municipalities. Source: authors’ own elaboration.

**Figure 4 ijerph-19-03359-f004:**
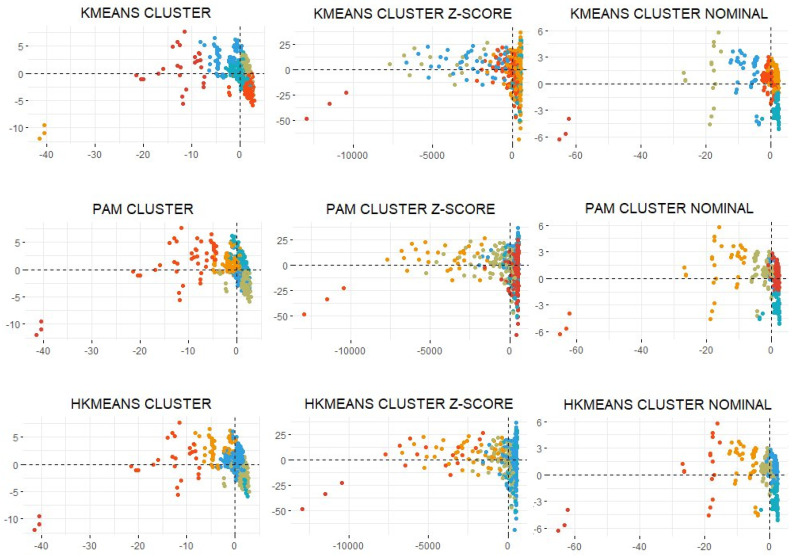
Representation of the results of the k-means, PAM, and hierarchical k-means algorithms for the different data sets. Source: authors’ own elaboration.

**Figure 5 ijerph-19-03359-f005:**
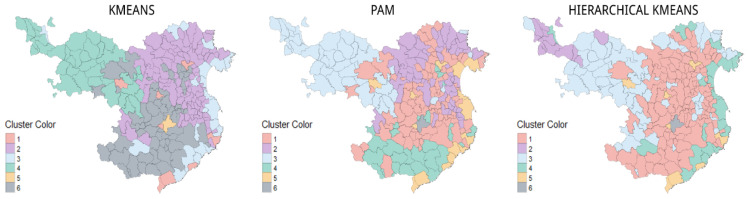
Representation of the cluster map: k-means, PAM and hierarchical k-means, according to the normal dataset, to observe their spatial distribution. Source: authors’ own elaboration.

**Figure 6 ijerph-19-03359-f006:**
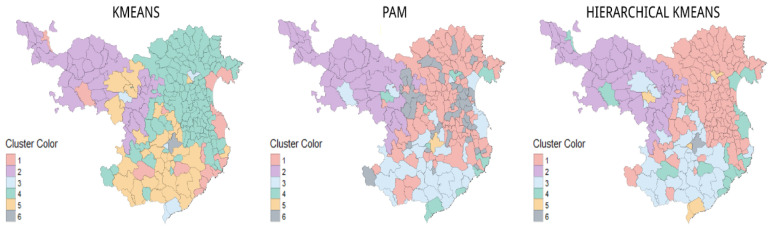
Representation of the cluster map: k-means, PAM, and hierarchical k-means, according to the nominal dataset, to observe their spatial distribution. Source: authors’ own elaboration.

**Figure 7 ijerph-19-03359-f007:**
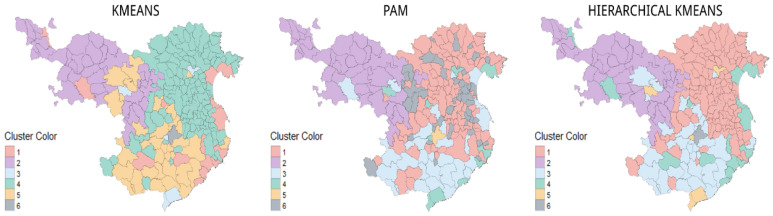
Representation of the cluster map: k-means, PAM, and hierarchical k-means, according to the z-score dataset, to observe their spatial distribution. Source: authors’ own elaboration.

**Table 1 ijerph-19-03359-t001:** Study of the number of optimal variables from different variable selection methods, according to the MSE.

Method	MSE	Number of Variables	
		Selected	Non-Selected
Ridge Regression	25,981.73	54	0
Lasso	55,404.96	12	42
Elastic Net	70,199.54	53	1
SCAD	50,711.94	14	40
MCP	50,711.94	16	38
LARS	41,167.40	34	17
Spike and Slab	25,302.36	53	1

Source: author’s own elaboration.

**Table 2 ijerph-19-03359-t002:** External and internal validation of clustering.

Name	Nº Clusters	Noise Point	Avg Between	Avg Within	Avg Silhouette	DUNN Index	Entropy	WB Ratio	CH Index	Separation Index
Data Set: Original										
*K-MEANS*	6	0	9.962	7.569	0.084	0.087	1.407	0.760	91.998	2.877
*PAM*	6	0	9.836	7.639	0.065	0.065	1.509	0.777	85.240	2.567
*CLARA*	6	0	10.499	8.070	0.074	0.038	0.961	0.769	59.973	2.488
*CLARANS*	6	0	10.064	7.766	0.070	0.068	1.206	0.772	83.459	2.739
*HKMEANS*	6	0	10.407	7.639	0.120	0.078	1.217	0.734	89.323	3.174
*FUZZY*	3	0	9.928	9.000	0.067	0.025	0.580	0.907	27.103	1.716
*BIRCH*	6	0	9.232	8.614	−0.073	0.029	1.671	0.933	18.810	2.437
*BICO*	6	0	9.560	8.539	−0.030	0.024	1.343	0.893	16.487	2.304
*EA*	6	0	9.560	8.539	−0.030	0.024	1.343	0.893	16.487	2.304
*DIANA*	4	0	12.017	9.128	0.024	0.044	0.256	0.760	6.536	3.020
*AGNES*	4	0	10.363	9.130	−0.072	0.044	0.422	0.881	5.193	2.886
*Data set: Nominal*										
*K-MEANS*	6	0	9.191	5.266	0.196	0.065	1.241	0.573	278.179	2.232
*PAM*	6	0	7.641	7.379	−0.101	0.011	1.509	0.966	3.012	1.006
*CLARA*	6	0	8.728	5.459	0.123	0.037	1.228	0.625	244.045	1.403
*CLARANS*	6	0	8.694	5.358	0.137	0.037	1.293	0.616	256.041	1.499
*HKMEANS*	6	0	9.195	5.268	0.195	0.065	1.240	0.573	278.081	2.241
*FUZZY*	4	0	9.109	6.255	0.077	0.015	0.862	0.687	141.029	1.567
*BIRCH*	6	0	7.609	6.744	−0.137	0.008	1.563	0.886	25.802	1.031
*BICO*	6	0	7.808	6.465	−0.008	0.012	1.517	0.828	26.727	1.259
*EA*	6	0	7.808	6.465	−0.008	0.012	1.517	0.828	26.727	1.259
*DIANA*	4	0	7.158	7.470	−0.089	0.012	0.243	1.044	0.789	1.440
*AGNES*	4	0	6.461	7.451	−0.233	0.008	0.422	1.153	1.680	1.165
*Data set: Z-score*										
*K-MEANS*	6	0	10.149	7.759	0.061	0.104	1.241	0.765	83.072	2.789
*PAM*	6	0	9.352	9.103	−0.039	0.036	1.509	0.973	3.456	2.374
*CLARA*	6	0	10.013	7.846	0.060	0.093	1.228	0.784	78.518	2.382
*CLARANS*	6	0	10.014	7.766	0.079	0.099	1.293	0.775	83.033	2.422
*HKMEANS*	6	0	10.15	7.758	0.061	0.104	1.240	0.764	83.097	2.771
*FUZZY*	4	0	10.263	8.413	0.038	0.049	0.862	0.820	65.039	2.406
*BIRCH*	6	0	9.266	8.711	−0.116	0.040	1.563	0.940	16.763	2.478
*BICO*	6	0	9.434	8.515	−0.028	0.039	1.517	0.903	20.023	2.400
*EA*	6	0	9.434	8.515	−0.028	0.039	1.517	0.903	20.023	2.400
*DIANA*	4	0	8.796	9.185	−0.087	0.051	0.243	1.044	1.643	3.160
*AGNES*	4	0	8.491	9.194	−0.156	0.039	0.422	1.083	1.554	2.841

Source: authors’ own elaboration.

**Table 3 ijerph-19-03359-t003:** Distribution of the number of cases according to the data set and the type of grouping.

Name	Cluster 1 (C1)			Cluster 2 (C2)			Cluster 3 (C3)			Cluster 4 (C4)			Cluster 5 (C5)			Cluster 6 (C6)		
	O ^1^	N ^2^	Z ^3^	O ^1^	N ^2^	Z ^3^	O ^1^	N ^2^	Z ^3^	O ^1^	N ^2^	Z ^3^	O ^1^	N ^2^	Z ^3^	O ^1^	N ^2^	Z ^3^
*K-MEANS*	25	44	127	258	127	15	62	15	3	121	360	360	3	114	114	194	3	4
*PAM*	235	235	235	165	117	117	122	97	97	92	30	30	46	3	3	3	181	181
*CLARA*	425	239	328	163	95	115	58	284	25	5	38	6	10	4	2	2	3	187
*CLARANS*	347	277	235	166	122	117	101	49	97	41	5	30	5	3	3	3	207	181
*HKMEANS*	355	360	360	41	132	132	183	109	109	57	44	44	24	15	15	3	3	3
*FUZZY*	170	345	597	492	33	34	1	1	1	0	284	28	0	0	3	0	0	0
*BIRCH*	44	32	32	96	43	44	161	219	170	50	132	53	126	50	128	186	187	236
*BICO*	95	198	193	35	48	101	6	3	3	331	179	122	151	74	153	45	161	91
*EA*	95	3	3	45	161	91	331	198	193	151	179	122	35	74	153	6	48	101
*DIANA*	627	630	612	24	18	33	9	12	15	3	3	3	0	0	0	0	0	0
*AGNES*	594	594	594	33	33	33	33	33	33	3	3	3	0	0	0	0	0	0

Source: authors’ own elaboration. ^1^: cluster based on the original data set. ^2^: cluster based on the nominal data set. ^3^: cluster based on the z-score data set. Source: authors’ own elaboration.

**Table 4 ijerph-19-03359-t004:** Measurement of the number of cases that vary between clusters to study the variability of results.

	*0 Changes*	*1 Changes*	*2 Changes*	*0 Changes*	*1 Changes*	*2 Changes*	*0 Changes*	*1 Changes*	*2 Changes*
	*Data Set: Original*			*Data Set: Nominal*			*Data Set: Z-Score*		
*K-MEANS*	202	19	0	217	4	0	217	4	0
*PAM*	120	98	3	74	142	5	74	142	5
*CLARA*	155	65	1	56	162	3	56	162	3
*CLARANS*	181	40	0	56	162	3	56	162	3
*HKMEANS*	196	25	0	217	4	0	217	4	0
*FUZZY*	54	167	0	10	210	1	10	210	1
*BIRCH*	172	46	3	197	24	0	197	24	0
*BICO*	172	46	3	196	25	0	173	48	0
*EA*	172	46	3	197	24	0	173	48	0
*DIANA*	172	46	3	197	24	0	221	0	0
*AGNES*	172	46	3	197	24	0	221	0	0

Source: authors’ own elaboration.

**Table 5 ijerph-19-03359-t005:** Descriptive analysis by conglomerates, only robust values (median (1st quartile–3rd quartile)).

*FRENCH BORDER (C1)*	*MOUNTAIN (C2)*	*INLAND (C3)*	*COASTAL (C4)*	*OTHERS (C5)*	*CAPITAL (C6)*	*FRENCH BORDER (C1)*	*MOUNTAIN (C2)*	*INLAND (C3)*	*COASTAL (C4)*	*OTHERS (C5)*	*CAPITAL (C6)*
*n = 360*	*n = 132*	*n = 109*	*n = 44*	*n = 15*	*n = 3*	*n = 360*	*n = 132*	*n = 109*	*n = 44*	*n = 15*	*n = 3*
pob_res_alestranger						cadastre_parcel_u					
12 (6–25)	10 (4–16)	35 (14–65.25)	324 (276–456)	855 (685–1263)	4160 (3941–4339.5)	407.5 (235.25–698.5)	407.5 (194.75–608.75)	1330.5 (957.5–2527.5)	4540 (2292–7232)	6541 (5850.5–9234)	10,649 (10,645–10,677)
saldo_migratori_intern						cadastre_inmo_u					
1 ((−7)–8.25)	1 ((−5)–6)	4.5 ((−8)–30.25)	−8 ((−42)–33)	−2 ((−30.5)–55)	9 ((−39.5)–51)	463.5 (247.25–878.5)	579.5 (242–1139)	2372.5 (1166–4260.75)	15,982 (8132–21,720)	32,691 (26,451.5–38,173.5)	79,579 (79,242–79,713.5)
saldo_migratori_extern						cadastre_valor					
2 (0–6)	1 (0–4)	8 (0–18)	48 (2–85)	3 ((−16.5)–212.5)	546 (310–699)	22,797.5 (13,956.5–44,019.5)	24,778.5 (11,902–70,339.75)	144,625.5 (64,595–250,763.25)	686,228 (312,990–12,862,52)	1,374,697 (1,291,656–2,110,633.5)	4005,166 (3,806,792.5–4,036,354.5)
saldo_migratori_total						atur_mig					
2 ((−5)–11)	3 ((−2.25)–7)	14.5 ((−4.5)–40.25)	22 ((−11)–107)	28 ((−6.5)–202.5)	555 (361–659.5)	22.71 (9.79–43.46)	10.5 (4.83–36.605)	124.915 (56.603–279.955)	625.17 (477.92–948.83)	2190.42 (1582.955–3230.75)	5730.42 (5447.835–6093.585)
irpf_base_imp						atur_mig_estranger					
20,129 (18,708.25–21,803.5)	19,582.5 (17,367–21,283.5)	20,578.5 (19,228.75–21,582.5)	18,577 (17,700–19,991)	18,736 (17,123–19,331.5)	24,800 (24,443–25,100)	2.96 (1.08–7.123)	0.96 (0.06–3.455)	15.54 (3.958–36.293)	184.33 (127.67–346.58)	619.08 (432.5–820.5)	1644.83 (1552.415–1774.29)
irpf_couta_auto						inde_env					
5129.5 (4538.25–5815)	4831.5 (4158–5676)	4745 (4381.25–5296)	4749 (4540–5065)	4513 (4197–4740.5)	6647 (6615.5–6733)	130.255 (101.812–157.438)	154.23 (119.182–192.27)	93.6 (82.613–117.955)	97.55 (88.81–120.07)	81.53 (77.885–116.02)	81.95 (81.38–85.05)
nascuts_vius						tax_bruta_mort					
4 (2–9)	3 (1–7)	27.5 (14–55.25)	100 (84–143)	302 (290–342.5)	1048 (1041.5–1074.5)	9.16 (6.455–12.795)	9.05 (5.695–13.413)	7.805 (6.412–9.773)	8.42 (7.85–9.32)	7.68 (6.265–8.95)	7.22 (7.215–7.42)
morts_num						index_rec					
2 (1–4)	1 (0.75–3)	13.5 (7–23.25)	42 (29–61)	131 (106–137.5)	344 (338.5–357)	147.22 (110–196.243)	158.57 (124.52–217.957)	109.7 (99.032–129.367)	107.47 (100.38–146.15)	108.23 (101.18–112.64)	96.05 (95.695–96.14)
saldo_pobl						index_dep_glob					
2 (0–5)	1 (0–4)	16 (6–29.25)	65 (40–88)	193 (161.5–208.5)	704 (684.5–736)	60.595 (54.788–64.713)	56.185 (49.905–62.543)	54.47 (52.33–56.37)	54.04 (52.64–54.87)	51.12 (42.165–52.085)	50.06 (49.785–50.24)
mobilitat_estudiants_uni_foramun						edat_mitja					
33.511 (5–20)	57.586 (0–11.25)	44.974 (40–105)	111.312 (135–300)	272.496 (597.5–660)	58.381 (1120–1177.5)	44.15 (42.2–45.725)	45.40 (43.6–47.2)	41.50 (40.275–43.225)	41.50 (40.8–43)	41.30 (39.9–42.5)	40.0 (39.9–40.1)
mobilitat_estudiants_uni_mun						creix_natu					
0 (0–0)	0 (0–0)	0 (0–0)	0 (0–0)	0 (0–0)	10,785 (10,737.5–10,890)	−1 ((−3)–1)	−1 ((−3)–1)	2 ((−4)–11.25)	12 (0–28)	3 ((−22.5)–120.5)	326 (313–336)
renda_mitja						index_sint_fecund					
12,195 (11,375.5–13,261.5)	13,084 (12,102.75–14,568.25)	12,304 (11,315.25–13,375.25)	10,629 (9818–11,665)	10,104 (9600.5–10,938.5)	13,183 (12,930.5–13,355)	1.28 (0.838–1.74)	1.3 (0.768–1.74)	1.415 (1.175–1.675)	1.45 (1.3–1.61)	1.350 (1.2–1.74)	1.45 (1.435–1.49)
total_pobl						taxa_estreng					
579 (284.75–1035.75)	340.5 (181.75–829)	3525.5 (1713.5–5474.5)	10,709 (10,231–17,677)	37,042 (33,972–39,096)	98,255 (97,920.5–98,634)	0.101 (0.066–0.138)	0.061 (0.049–0.112)	0.082 (0.044–0.117)	0.217 (0.164–0.298)	0.225 (0.161–0.258)	0.18 (0.179–0.182)
biblio						index_autoc					
0 (0–0)	0 (0–0)	1 (0–1)	1 (1–3)	3 (1–5)	18 (18–18)	30.685 (24.788–34.858)	35.03 (24.377–40.385)	28.525 (23.395–35.197)	33.11 (21.49–38.24)	32.88 (20.445–37.505)	40.22 (40.175–40.22)
ss_total_mig						densitat_pob					
228 (113–405)	145 (83–326)	1539 (793–2293)	4109 (3671–6547)	13,633 (12,460–14,595)	39,427 (38,747–40,147)	41 (20–76)	16 (5–34)	135 (60.75–190)	423 (175–630)	1171 (758.5–2214)	2512 (2503.5–2521.5)
ss_ext_mig						contract_tempo					
9.83 (6.332–14.315)	5.42 (2.015–9.15)	7.535 (3.947–11.123)	17.08 (14.14–22.09)	17.37 (13.51–24.36)	16.77 (16.405–17.125)	0.812 (0.5–1)	0.883 (0.702–1)	0.834 (0.75–0.906)	0.838 (0.794–0.886)	0.861 (0.819–0.902)	0.898 (0.893–0.9)
ss_agricultura_per						gini					
6.606 (3.541–12.228)	7.23 (2.91–12.821)	2.61 (1.487–5.697)	2.572 (1.55–4.059)	1.277 (0.384–2.425)	0.627 (0.596–0.633)	31.3 (28.8–33.6)	31.9 (28.975–34.8)	28.5 (27.4–30.6)	34.6 (32.7–36.1)	34.1 (31.7–36.6)	36 (35.45–36.1)
ss_industria_per						renda_bruta_mitja					
11.765 (8.747–16.981)	15.155 (6.744–23.149)	20.977 (15.936–28.685)	9.818 (7.502–14.758)	11.207 (10.869–19.345)	12.768 (12.647–12.857)	14,791 (13,581.5–16,171.5)	15,874 (14,383.75–17,778.25)	14,926.5 (13,472.5–16,295.75)	12,626 (11,634–13,970)	12,011 (11,342.5–12,968)	16,303 (16,006.5–16,559)
ss_construccio_per						renda_salari					
8.889 (6.589–10.714)	7.833 (5.66–10.086)	7.93 (6.58–9.378)	9.756 (7.456–10.343)	6.298 (5.141–6.65)	4.661 (4.655–4.77)	8258.5 (7528–9185)	8732.5 (7775–10,044.5)	9430 (8399–10,639)	7393 (6793–8117)	7218 (6956–7662.5)	10,277 (10,067.5–10,454.5)
ss_serveis_per						renda_pensions					
70.588 (64.057–74.803)	67.458 (60.34–75.506)	65.896 (61.232–72.396)	75.795 (65.677–78.832)	79.983 (69.048–80.24)	81.937 (81.779–82.07)	2861 (2546–3379.75)	3209 (2814.75–3747.25)	2717 (2463.75–2959.75)	2488 (2174–2744)	2221 (1795–2749.5)	2963 (2920.5–3007)
equipament						renda_atur					
0 (0–3.978)	0 (0–6.082)	2.61 (1.768–4.425)	2.06 (1.32–2.78)	0.81 (0.44–2.05)	1.83 (1.825–1.835)	237.5 (189.75–294)	234.5 (184.75–287)	242.5 (209–283.5)	326 (282–358)	305 (255.5–401.5)	245 (235–263.5)
preu_mig_lloguer						capitalcomarca					
487.73 (432.805–522.745)	472.56 (387.272–514.478)	498.545 (435.03–545.448)	454.62 (408.96–480.18)	422.2 (378.66–434.205)	515.46 (500.545–538.245)	0 (0–0)	0 (0–0)	0 (0–0)	0 (0–1)	0 (0–1)	1 (1–1)
num_habitatges						geo_altitud					
6 (3–13)	4 (1.75–11.25)	34.5 (16.75–78.25)	190 (141–296)	842 (712.5–917)	3267 (3199–3291.5)	82 (33.75–161)	953.5 (362–1180.5)	111 (89.75–172)	31 (12–148)	39 (13–260)	70 (70–70)
transit_victim						munt					
111.5 (1–280.25)	121 (1–298.5)	132 (2–260.5)	111 (1–263)	92 (2–263.5)	76 (38.5–186.5)	0 (0–0)	0 (0–0)	0 (0–0)	1 (0–1)	0 (0–1)	0 (0–0)
trucades_emer						costa					
2 (1–6)	2 (1–5)	3 (1–10)	5 (2–23)	5 (1.5–12.5)	1 (1–253.5)	0 (0–0)	1 (1–1)	0 (0–0)	0 (0–0)	0 (0–1)	0 (0–0)
super_conreu_herb						latitud					
15 (4–67.5)	10.5 (3–23.75)	10 (3–21)	36 (9–102)	4 (2–30.5)	3 (3–9.5)	42.175 (42.038–42.298)	42.257 (42.144–42.35)	41.935 (41.827–42.03)	42.125 (41.917–42.219)	42.182 (41.699–42.237)	41.982 (41.982–41.982)
super_conreu_lleny						longitud					
110.5 (0–279.25)	120 (0–297.5)	131 (0–259.5)	110 (0–262)	91 (0–262.5)	75 (37.5–185.5)	2.946 (2.812–3.04)	2.327 (2.072–2.612)	2.76 (2.638–2.883)	3.073 (2.662–3.129)	2.792 (2.657–2.848)	2.824 (2.824–2.824)

**Table 6 ijerph-19-03359-t006:** Probability of a municipality belonging to each of the clusters (odds ratio).

	French Border	Mountain	Inland	Coastal	Others
*Intercept*	0.9992802 (***)	1.0002685 (***)	0.9998384 (***)	1.0004043 (***)	1.0006086
*Population residing abroad*	1.0083055 (***)	1.0002468 (*)	0.9887537 (***)	1.0011125 (***)	0.9988869
*Internal migratory balance*	0.9902796 (***)	0.9960504 (***)	1.0013632 (***)	0.9811316 (***)	0.9980404
*External migratory balance*	1.0023456 (***)	0.9995994 (***)	0.995847 (***)	0.9994775 (***)	0.999062
*Taxable base of personal income tax*	0.999858	0.999797	1.0000104	1.0009097 (**)	0.9995569
*Self-employed income tax contributions*	1.000437	1.00045	1.000327	1.000503	1.000133
*Number of births*	0.9866577 (***)	1.0149668 (***)	0.9957985 (***)	1.0328247 (***)	0.9798185
*Number of deaths*	0.9873149 (***)	1.0627083 (***)	1.0011853 (***)	1.0125107 (***)	0.9523698
*Population balance*	0.9993343 (***)	0.9550756 (***)	0.9946195 (***)	1.020063 (***)	1.0288215
*Mobility of university students outside the municipality*	0.9887809 (***)	0.9835925 (***)	1.0066694 (***)	1.0053018 (***)	1.0113427
*Mobility of university students within the municipality*	1.0010445 (·)	0.9985847 (***)	1.0020055 (***)	0.9969711 (***)	0.9982406
*Average income*	0.9991602 (*)	0.9999274	0.9992583 (·)	0.9990069 (·)	0.9996053
*Total population*	0.9979256 (***)	1.0011634 (***)	0.9986322 (***)	0.9928825 (***)	0.9990906
*Library count*	0.9920155 (***)	0.9969202 (***)	1.0082681 (***)	1.0105826 (***)	0.9944124
*Average number of people registered with social security*	1.0036874 (***)	0.9979587 (**)	1.0018299 (**)	1.0142032 (***)	1.0004368
*Average number of foreigners registered with social security*	0.9832582 (***)	0.9783802 (***)	1.0052646 (***)	1.0045112 (***)	1.0420857
*Percentage of workers engaged in the agricultural sector registered with social security*	0.9919438 (***)	0.9796335 (***)	0.981117 (***)	1.0864516 (***)	0.9754141
*Percentage of workers in industry registered with social security*	0.9765902 (***)	1.0136796 (***)	1.0162364 (***)	0.9900073 (***)	1.0222547
*Percentage of workers in the construction sector registered with social security*	1.0616384 (***)	1.0078621 (***)	0.9817759 (***)	0.9884191 (***)	0.9674953
*Percentage of workers in the services sector registered with social security*	1.0120162 (***)	0.9976513 (***)	0.9992377 (***)	0.9597444 (***)	1.0410581
*Sports facilities count*	0.9866832 (***)	1.0077482 (***)	0.9896743 (***)	1.0081863 (***)	1.0040921
*Average rental price*	0.9997722	1.0005119	1.0012895 (·)	0.9906604 (***)	0.9999714
*Count of homes available for rent*	0.9943651 (***)	0.9992177 (***)	0.9999074 (·)	0.9828655 (***)	1.0073002
*Emergency calls count*	0.9970014 (***)	1.0067798 (***)	1.0035279 (***)	1.015406 (***)	1.0054075
*Area of woody cultivation*	1.0163713 (***)	0.9976557 (***)	0.9898313 (***)	1.0010048 (·)	0.9971145
*Number of properties according to land register*	1.0001438	0.9999955	1.0003972 (·)	1.0017136 (***)	1.0002099
*Average unemployment*	1.007197 (***)	1.001434 (***)	1.006958 (***)	1.005407 (***)	1.004737
*Aging ratio*	0.9945135 (***)	0.9960614 (***)	0.9962998 (***)	0.9685533 (***)	0.9920995
*Active population replacement rate*	1.0006615	0.9971274 (**)	0.9963596 (***)	0.9999958	1.0030382
*Middle age*	1.0023732 (***)	1.0010506 (***)	0.9681195 (***)	1.0175038 (***)	1.0253728
*Synthetic fertility rate*	0.9998299 (***)	0.9925758 (***)	0.999746 (***)	1.0019436 (***)	1.0064768
*Proportion of native born population*	0.9526581 (***)	0.993467 (***)	1.0901165 (***)	1.0573994 (***)	0.9445746
*Percentage of the number of temporary contracts*	1.004969 (***)	0.9980301 (***)	0.9977705 (***)	1.002199 (***)	0.9973226
*Average gross income*	1.0006414 (*)	1.0004035	1.0006079 (·)	0.9984342 (**)	1.0003484
*Average income from pensions*	1.0004604	1.000368	0.9998378	1.0016645 (**)	1.0009924
*County capital (no)*	0.9981343 (***)	0.9978039 (***)	0.9988672 (***)	1.007165 (***)	0.9986435
*Municipality located in the mountains (no)*	1.0027483 (***)	1.0010293 (***)	0.9967967 (***)	1.0005993 (***)	0.9989383
*Latitude*	0.9739104 (***)	1.011893 (***)	0.9863013 (***)	1.0184511 (***)	1.0271034
*Number of traffic deaths*	0.9848847 (***)	1.0027088 (***)	1.0106366 (***)	0.9989089 (·)	1.0035241
*Area for herbal cultivation*	1.0016808 (*)	0.9997817	0.9872208 (***)	1.0102234 (***)	1.0029894
*Number of plots according to land register*	1.0003769	1.0006836	1.0015421 (**)	0.9990165 (*)	1.000732
*Total cadastral value*	0.999998	0.9999899 (*)	0.9999944	0.9999917 (*)	1.0000028
*Average foreign born unemployment*	1.0109255 (***)	0.9937618 (***)	0.9975241 (***)	1.0235579 (***)	0.9959258
*Gross mortality rate*	1.0067221 (***)	1.0322658 (***)	0.9621324 (***)	1.0172427 (***)	0.9872505
*Overall dependency ratio*	1.0779853 (***)	0.9850141 (***)	0.9621172 (***)	1.0158033 (***)	0.9834363
*Natural population growth*	0.9792645 (***)	1.0495036 (***)	0.9959008 (***)	0.9769135 (***)	0.9939988
*Immigration rate*	0.9994612 (***)	0.999584 (***)	0.9999851 (***)	1.0002735 (***)	1.0007975
*Population density*	0.9987993 (*)	0.9982249 (**)	0.9990685 (·)	0.9983053 (*)	0.9988891
*Gini index*	0.9536537 (***)	1.0278178 (***)	0.9691616 (***)	1.013044 (***)	1.048768
*Average income from salary*	1.0003585	0.9998022	1.0004226	1.000985 (·)	1.0003853
*Average income from unemployment benefits*	1.0049536 (***)	1.0021693 (**)	1.0039937 (***)	1.0138551 (***)	0.9964328
*Altitude*	0.9969976 (***)	1.0022741 (***)	0.9995159	1.0034163 (***)	0.9988485
*Municipality located on the coast (no)*	0.9786072 (***)	1.0144727 (***)	0.9987252 (***)	1.0047179 (***)	1.0041124
*Length*	1.002225 (***)	0.9997354 (***)	0.9968903 (***)	1.0015902 (***)	1.0005632

(***) = *p* ≤ 0.001. (**) = *p* ≤ 0.01. (*) = *p* ≤ 0.05. (·) = *p* ≤ 0.1. Source: authors’ own elaboration.

## Data Availability

All the data, including the code to produce the figures, can be requested from the first author (xperafita@dipsalut.cat).

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
