# Peer review of "Clustering of Small Territories Based on Axes of Inequality"

_ijerph, 2022, doi:10.3390/ijerph19063359_

Round 1

Reviewer 1 Report

1.Literature Review part should be improved which could not only includes the advantages of e-cohorts, but also should reviews how different variables or groups influences inequality in past research. 

2. The design research should be improved, explain why and how to select 54 variables, it better use a paragraph to introduce and explain(or use a frame pic.) 

3.The discussion part looks ok, but the conclusion part is too simple,should be supplemented.

Author Response

Reviewer # 1

We thank the reviewer for all their comments.

1.Literature Review part should be improved which could not only includes the advantages of e-cohorts, but also should reviews how different variables or groups influences inequality in past research.

Although we believe that what the reviewer has requested goes far beyond the objective of our study, we have expanded the introduction as suggested by the reviewer.

On pages 1 and 2,

‘1.-. Background

Currently, the concept of 'health inequalities' refers to the impact that factors such as wealth, education, employment, racial or ethnic group, exposure to environmental factors such as air pollution or weather variables, urban or rural residence, and/or the social conditions of an individual’s workplace or dwelling has on the distribution of health and disease among the population. The study of the characteristics of the population and the geographical area of ​​residence is the methodological support that allows for intervention points focused on the prevention and the disappearance of existing health inequalities to be identified.

In its origin, socioeconomic inequalities were identified with inequality in health [1]. Health inequality can be defined as an inequity in the spread of the disease. In other words, health inequality is the systematic and potentially avoidable differences in one or more health aspects across socially, economically, demographically, or geographically defined populations or population groups. Two conditions must be met for a difference in health to be considered an inequality: 1) it must be considered socially unjust and 2) be potentially avoidable (i.e., there are instruments available that could be used to avoid it) [1].

There is evidence that inequalities in health exist. While the Ladonde [2] and Black [3] Reports pointed this out, it would be the Acheson Report [1] that firmly concluded that inequalities in health have a socioeconomic explanation. Today, twenty years later, most of these relationships have been demonstrated, and a not insignificant proportion of them are caused by environmental problems [1]. These factors are generally, but not exclusively, linked to gender, social and economic conditions [1,4,5].

In general, the living environment, and thus environmental conditions, can contribute to socioeconomic inequalities in health either independently or, more likely, jointly [1,5]. The first is differential exposure: the most economically disadvantaged groups have a greater exposure to environmental problems including, among others, air pollution. The second is differential susceptibility to exposure, (i.e., the main adverse health effects), resulting from the environmental problems, which occur in the most economically disadvantaged people due to their greater vulnerability.

When we think about a longitudinal study to see how health inequalities, individuals’ health, income, or other specific characteristics, evolve over time, our thoughts very quickly turn to creating a cohort. This is immediately (…)’

  1. The design research should be improved, explain why and how to select 54 variables, it better use a paragraph to introduce and explain (or use a frame pic.) 

Following the reviewer’s indications, we carried out a review of all the existing indicators found in the main databases in the province of Girona and from which we obtained 541 variables. These were then processed based on the availability of data for the study period, data availability for most municipalities throughout the study period, as well as the elimination of variables that we have considered to be duplicates or redundant, and those that did not contribute any relevant information for the study.

This, on page 5,

‘2.1.-. Methods prior to carrying out the study, the data set, and the data sources

As explained earlier, the diversity of the territory of Girona requires a large number of variables to determine the differences and similarities between its municipalities. These differences can be from an economic point of view, where the main cities in the province have a larger number of specific companies and sectors, to the migratory processes that the areas experience or the number of elderly people who live there. As can be seen in Figure 1, we carried out a review of all the indicators that exist in the main databases that provide information on the municipalities in the province of Girona. From this, we obtained 541 variables. These were then processed based on the availability of data for the study period, data availability for most municipalities throughout the study period, as well as the elimination of variables that we have considered to be duplicates or redundant, and those that did not contribute any relevant information for the study.

Prior to the clustering, a final set of 54 potential variables encompassing (…)’

We have also added a figure (new Figure 1) in which we explain the process of debugging all detected variables up to the final data model.

3.The discussion part looks ok, but the conclusion part is too simple,should be supplemented.

As the reviewer suggested, we have expanded the conclusions by adding four paragraphs on: what the article seeks to demonstrate, possible limitations with the interaction of the future cohort with special data, and the need to generate data at the individual level or below municipal needs to better show inequality.

On pages 4 and 5 of 27,

‘5.-. Conclusions

This article aims to help researchers and other decision-making institutions facilitate a comparison of the structuring and grouping of small areas, especially in those cases where the differences between them are so large. It also endeavours to show an optimal way of transforming and working on datasets to facilitate the resulting groupings. Two of the main limitations in grouping such diverse and small populations is, on the one hand, the lack of data and, on the other, the lack of experiences that have endured over time where we can observe their evolution.

If we want to analyse the impacts of spatial variables such as NDVI or pollutants PM2.5, PM10, NO2 or CO2, it is advisable to generate data at a lower level than the municipality, as municipalities while not the smallest administrative division, are the smallest division that has political decision-making power. This would allow us to segment a cohort from census tracts or districts in the future and reduce the potential ecological fallacies that cohort data may generate. In addition, it would better capture the inequality that can be observed between the rich and poor areas in cities. The lack of experience working in small areas, along with the nature of most indicators, makes these processes difficult.

Currently, it is essential to start generating data at the scale of small areas, even smaller than those of a municipality, because otherwise we will not always be masking inequalities through averages and aggregate values ​​of population subsets where wealth has blurred the levels of poverty. On the other hand, the microdata will permit the creation and adaptation of new indicators that allow the inequalities and the phenomena that occur in the territorial field to be captured more efficiently.

Data protection policies, although necessary, often prevent the study of the reality of territories. They also make it difficult to study individuals in a particular way. These mechanisms end up making it difficult to observe inequalities as well as study the sensitivity that each individual has with respect to their social conditions and how these affect them.

To facilitate the best clustering process (…)’

Reviewer 2 Report

1. Please stress precisely what is the novelty here for environmental science (statistical methods are known and standard).

2. Where is the information about sample size and its relationship with tests?

3. In p.8 you have symbol "g", what is it? And why j starts with 2?

4. You did not discuss statistical assumptions behind any method. Why do you choose non-parametric tests? What about the assumptions of multinomial logistic regression?

5. Provide please more detailed captions to all tables and figures and labels for all axis.

6. I would recommend representing the data in tables concise way
(rounding, there is also no need to include big tables in the central part of the article as it does not shed light on the main context.)

Author Response

Reviewer # 2

We thank the reviewer for all their comments.

  1. Please stress precisely what is the novelty here for environmental science (statistical methods are known and standard).

We have tried to respond to this comment by expanding the conclusions with four new paragraphs on: what the article seeks, possible limitations with the interaction of the future cohort with special data, and the need to generate data at the individual level or below municipal needs to better show inequality.

On pages 4 and 5 of 27,

‘5.-. Conclusions

This article aims to help researchers and other decision-making institutions facilitate a comparison of the structuring and grouping of small areas, especially in those cases where the differences between them are so large. It also endeavours to show an optimal way of transforming and working on datasets to facilitate the resulting groupings. Two of the main limitations in grouping such diverse and small populations is, on the one hand, the lack of data and, on the other, the lack of experiences that have endured over time where we can observe their evolution.

If we want to analyse the impacts of spatial variables such as NDVI or pollutants PM2.5, PM10, NO2 or CO2, it is advisable to generate data at a lower level than the municipality, as municipalities while not the smallest administrative division, are the smallest division that has political decision-making power. This would allow us to segment a cohort from census tracts or districts in the future and reduce the potential ecological fallacies that cohort data may generate. In addition, it would better capture the inequality that can be observed between the rich and poor areas in cities. The lack of experience working in small areas, along with the nature of most indicators, makes these processes difficult.

Currently, it is essential to start generating data at the scale of small areas, even smaller than those of a municipality, because otherwise we will not always be masking inequalities through averages and aggregate values ​​of population subsets where wealth has blurred the levels of poverty. On the other hand, the microdata will permit the creation and adaptation of new indicators that allow the inequalities and the phenomena that occur in the territorial field to be captured more efficiently.

Data protection policies, although necessary, often prevent the study of the reality of territories. They also make it difficult to study individuals in a particular way. These mechanisms end up making it difficult to observe inequalities as well as study the sensitivity that each individual has with respect to their social conditions and how these affect them.

To facilitate the best clustering process (…)’

  1. Where is the information about sample size and its relationship with tests?

Unfortunately, we disagree with the reviewer, as we believe that we had already defined it this in the previous version of the manuscript.

In this sense,

In Section 2.-. Methods defined all the variables and explained the number of small areas as well as the transformation we applied to the three datasets.

In Section 3.1.-. Area and period of study, we defined the temporal dimension of the study.

In Table 3, we indicated the number of observations for each cluster and data set.

In Table 5, we showed the number of cases in each cluster.

However, that said, we have also attempted to make some clarifications in the text.

On page 8,

‘A z-score transformation was performed for the smoothed data set [46]. The same number of variables have been maintained in both datasets.’

On page 11,

‘The dimensions of the final dataset are defined in 54 variables for 221 municipalities over three years. Thus, obtaining a final sample of 35,802 cases’

  1. In p.8 you have symbol "g", what is it? And why j starts with 2?

We apologise for the mistake and this has now been changed in the new version of the manuscript.

On pages 9 and 10,

‘2.5.-. Data analysis

A multinominal logistic regression was carried out where the dependent variable () is the cluster generated, where j=1,2,3,4,5. The variable of the reference group was 6, modelled in the following way:

It was adjusted as follows to find the estimated probability () of the events:

The final result enables the clusters to be compared with the municipality of Girona’

  1. You did not discuss statistical assumptions behind any method. Why do you choose non-parametric tests? What about the assumptions of multinomial logistic regression?

We have added several sentences in the new version of the manuscript to remedy this.

On page 1 of 27,

‘3.6.-. Inference

Clusters represent the variability of the territory which, as we have shown, is very varied and therefore these different realities will be so different that they will not follow a normal distribution. The Kruskal-Wallis tests [68] and the Mann-Whitney tests [69] show that there are significant differences among the clusters. To see these differences through clusters, we assume that we do not have the presence of multiculturalism or outliers. A multinominal logistic regression was performed to see these differences [70]. The odds ratios of the regression are presented in Table 6.’

  1. Provide please more detailed captions to all tables and figures and labels for all axes.

We have followed the reviewer's suggestions in the new version of the manuscript.

On page 11,

Table 1. - Study of the number of optimal variables from different variable selection methods according to

the MSE.

Figure 2. - Process of obtaining the optimal number of clusters from the Elbow method

On page 13,

Table 3. - Distribution of the number of cases according to the data set and the type of grouping.

On page 14,

Figure 3. - Representation of the different algorithms performed to study the clustering of the municipalities

On page 15,

Figure 4. - Representation of the results of the K-MEANS, PAM and HIEARCHICAL KMEANS algorithms for the different data sets.

Figure 5. - Representation of the cluster map: K-MEANS, PAM and K-MEANS HIERARCHICAL according to the normal dataset to observe their spatial distribution

On page 16,

Figure 6. Representation of the cluster map: K-MEANS, PAM and K-MEANS HIERARCHICAL according to the nominal dataset to observe their spatial distribution

On page 17,

Table 4. - Measurement of the number of cases that vary from cluster to study the variability of results

On page 18,

Table 5. - Descriptive analysis by conglomerates, only robust values (Median (1st Quartile - 3rd Quartile)).

On page 1 of 27,

Table 6. - Probability of a municipality belonging to each of the clusters (ODDS Ratio).

  1. I would recommend representing the data in tables concise way
    (rounding, there is also no need to include big tables in the central part of the article as it does not shed light on the main context.)

We have followed the reviewer's recommendations.

In addition we have added a new table (Table 5).

However, we have not touched the odds ratio decimals because without them the differences between them would not be able to be seen.

Round 2

Reviewer 1 Report

The updated manuscipt is clearer now after serious extensions and revisions . I think it could be published after checking English writing style.